# Magnetic Solid-Phase Extraction of Cadmium Ions by Hybrid Self-Assembled Multicore Type Nanobeads

**DOI:** 10.3390/polym13020229

**Published:** 2021-01-11

**Authors:** Gabriela Buema, Adrian Iulian Borhan, Daniel Dumitru Herea, George Stoian, Horia Chiriac, Nicoleta Lupu, Tiberiu Roman, Aurel Pui, Maria Harja, Daniel Gherca

**Affiliations:** 1National Institute of Research and Development for Technical Physics, 47 Mangeron Boulevard, 700050 Iasi, Romania; gbuema@phys-iasi.ro (G.B.); aborhan@phys-iasi.ro (A.I.B.); dherea@phys-iasi.ro (D.D.H.); gstoian@phys-iasi.ro (G.S.); hchiriac@phys-iasi.ro (H.C.); nicole@phys-iasi.ro (N.L.); troman@phys-iasi.ro (T.R.); 2Faculty of Chemistry, Alexandru Ioan Cuza University of Iasi, 11, Carol I Boulevard, 700506 Iasi, Romania; aurel@uaic.ro; 3Integrated Center of Environmental Science Studies in the North Eastern Region—CERNESIM, Alexandru Ioan Cuza University of Iasi, Carol I nr. 11 Boulevard, 700506 Iasi, Romania; 4Faculty of Chemical Engineering and Environmental Protection, “Gheorghe Asachi” Technical University of Iasi, 73 Dimitrie Mangeron Street, 700050 Iasi, Romania

**Keywords:** adsorption, cadmium ions, magnetic nanobeads, one-pot synthesis, self-assembled ferromagnet-biopolymer

## Abstract

Novel hybrid inorganic CoFe_2_O_4_/carboxymethyl cellulose (CMC) polymeric framework nanobeads-type adsorbents with tailored magnetic properties were synthesized by a combination of coprecipitation and flash-cooling technology. Precise self-assembly engineering of their shape and composition combined with deep testing for cadmium removal from wastewater are investigated. The development of a single nanoscale object with controllable composition and spatial arrangement of CoFe_2_O_4_ (CF) nanoparticles in carboxymethyl cellulose (CMC) as polymeric matrix, is giving new boosts to treatments of wastewaters containing heavy metals. The magnetic nanobeads were characterized by means of scanning electron microscopy (SEM), powder X-ray diffraction analysis (XRD), thermogravimetric analysis (TG), and vibrational sample magnetometer (VSM). The magnetic properties of CF@CMC sample clearly exhibit ferromagnetic nature. Value of 40.6 emu/g of saturation magnetization would be exploited for magnetic separation from aqueous solution. In the adsorptions experiments the assessment of equilibrium and kinetic parameters were carried out by varying adsorbent dosage, contact time and cadmium ion concentration. The kinetic behavior of adsorption process was best described by pseudo-second-order model and the Langmuir isotherm was fitted best with maximum capacity uptake of 44.05 mg/g.

## 1. Introduction

Pollution with heavy metals (cadmium, copper, nickel, lead, chromium) from different industrial wastewaters represents a serious problem for the environment and human health [1]. Nanotechnology addresses the continuous development of solutions to the existing environmental problems and preventive measures for future problems. Nanoscience developments facilitate a number of emerging technologies to be addressed to solve the multiple problems of water in order to ensure the environmental stability and finally assisting the attainment of water quality standards and health advisories [2]. The development of cost-effective and stable materials, strategies and technology for providing pure water is a critical need for environmental protection. A number of technologies such as chemical precipitation, coagulation, reverse osmosis, membrane filtration, electrochemical reduction, ion exchange and adsorption are proposed [2,3,4]. One of the most used technologies to remove heavy metals from wastewater is adsorption [5,6,7,8,9,10,11,12]. An optimal adsorbent for this use should have a high surface area, high adsorption capacity and mechanical stability and be easily regenerated [13]. Cadmium is a toxic element that has been identified as a human carcinogen and teratogen impacting the lungs, liver and kidneys [14,15]. It is discharged into the environment from a multitude of sources: metal plating, metallurgic alloying, ceramics, the textile printing industry, photograph development, electroplating, alkaline battery manufacturing industries, smelting of non-ferrous metal ores, fossil fuel combustion and municipal waste incineration [16,17,18,19]. The US Environmental Protection Agency (USAEPA) has established a maximum contaminant level of 5 µg/L for cadmium in drinking water, while the World Health Organization (WHO) has set a maximum guideline concentration of 3 µg/L [20]. To date, several adsorbents, including chitosan-Fe_2_O_3_ [21], sSilica/Fe_3_O_4_ [22], activated carbon/Fe_3_O_4_ [23], magnetic alginate activated carbon (MAAC) beads [24], graphene/MnFe_2_O_4_ [25] and magnetic ferrite [26], have been considered for the removal of cadmium ions due to their unique properties such as high surface to volume ratio, surface functionalization, biocompatibility, reversibility and a comparatively low cost. However, there are certain limitations where the separation may require large external magnetic fields if the magnetic iron oxides embedded into composites are too small in size. The use of carbon-based nanostructures (CTN) as effective adsorbents is limited by the concerns raised on the potential toxicity of CNTs and their effect on human health and environment and the limited selectivity to various adsorbents [27]. Polymeric adsorbents have also been used for heavy metal adsorption applications, due to their adsorptive properties such as good pore sizes and flexible functional groups that widen their selectivity. However, their uses still have some important limitations, such as inadequacies in molecular structure, low selectivity to the target adsorbate and recycling issues [28]. To overcome these limitations, various organic-inorganic hybrid polymers have been used for the removal of heavy metals from wastewater [29,30,31,32,33]. In these composites, the functional variation of organic materials is combined with advantages of a thermally stable and robust inorganic substrate, resulting in strong binding affinities toward selected metal ions and relatively high metal ion adsorption capacities. These kinds of materials often present the best properties of each of its components in a synergic way. In recent years, there is an increasing interest in magnetic ferrite-based adsorbents due to their high adsorption capacities and unique magnetic properties for easy separation [26,32]. Magnetic nanoparticles (MNPs) formed from iron, cobalt, or nickel oxides show certain properties, including high surface-to-volume ratio and high magnetic moment, allowing potential manipulation by an external magnetic field [34].

Spinel cobalt ferrite (CoFe_2_O_4_) nanoparticles have captured researchers’ attention due to its high coercivity (Hc), anisotropy constant (K_1_) and saturation magnetization (Ms) values, chemical and thermal stability [35,36,37], easy preparation and rapid separation [38]. Researchers have proposed a considerable number of chemical/physical techniques for the synthesis of CoFe_2_O_4_ nanoparticles, comprised of sonochemical reactions [39], mechanical alloying [40], hydrothermal techniques, co-precipitation [36], micro-emulsion routes and sol gel processes [38,41,42,43]. Conventional preparation by coprecipitation methods requires the heating of the mixture up to 90 °C [44,45,46] or autoclave temperatures [47]. In addition, these products must be annealed at temperatures between 500 and 1100 °C to be crystallized [48,49,50]. For the specific preparation of magnetic ferrite for heavy metal removal from wastewater, it is necessary that the operating temperatures be above 60 °C [51,52]. The properties of CoFe_2_O_4_ nanoparticles, differently from bulk samples, are closely associated to the cation distribution, size, and shape. Cobalt ferrite has a partially inverse spinel structure with formula (Co_1−α_Fe_α_)_A_ [Co_α_Fe_2−α_]_B_O_4_, where A, B represent tetrahedral and octahedral sites, respectively. The notation α is inversion factor that ranges from zero to one for normal and invers spinel structures [53,54]. Other factors which may affect the physical properties are the quantum size and surface effects [55]. Magnetic studies demonstrated that the synthesis of CoFe_2_O_4_ nanoparticles by coprecipitation or thermal treatment of precursors led to a relative decrease of the saturated magnetization with decreasing particles size down to σs ~56–58 emu/g for d ~15 nm. The critical size for the transition to the superparamagnetic state appears to be ~5 nm around room temperature as a consequence of the strong magneto-crystalline anisotropy, preserved also for small nanoparticles [36,56,57]. At present, most of the magnetic nanoparticles may have possible toxic effects on the treated water. In order to avoid such problems, greener methods for the synthesis of these nanoadsorbents could be studied. Thus, nanotechnology plays a major role in performance improvement through the development of innovative methods to produce new products, to substitute existing production equipment and to reformulate new advanced nanomaterials resulting in less consumption of energy and materials and reduced harm to the environment as well as environmental remediation. In this perspective, inorganic-organic hybrid materials are considered to be of great importance in the development of future oriented advanced nanocomposite materials.

The objective of this paper is to contribute to such research development, supporting the research in functional materials by the opportunity to create a new smart and low toxic materials from inorganic and organic components, and the possibility of their assembly using nanostructured phases. In particular, the aim of present paper is to evaluate the feasibility of a novel hybrid inorganic CoFe_2_O_4_/CMC nanoparticle polymeric framework nanobeads-type adsorbent as an efficient technology for removing heavy metal ions from water. One can envisage a large potential for the use of such host organic polymers due to their multifunctionality derived from the combination of catalytically active oxide nanoparticles, but also due to their prompt processability, with coating forming properties. At the same time the innovation brought by hybrid nanocomposite formulation is intended to prevent any nanoparticles leaching into water, thus strongly limiting the potential threat associated with the dispersion of NPs into the environment. Herein, we have exploited a one-pot experimental method of self-assembled colloidal CoFe_2_O_4_ multicore nanobeads using carboxymethyl cellulose (CMC) as a polymeric framework. The present synthetic strategy for self-assembly involves: (i) the synthesis of CoFe_2_O_4_ via a coprecipitation solution phase colloidal technique and (ii) the assembly of the magnetic nanobeads by crash-cooling in ice water. This resulted in magnetic nanobeads with a densely packed CoFe_2_O_4_ multicore surrounded by a CMC polymer shell.

## 2. Materials and Methods

### 2.1. Reagents

The chemical reagents used in the present study were all of analytical grade and used without any additional purification steps. Solutions of iron (III) chloride and cobalt (II) chloride were prepared by dissolving FeCl_3_·6H_2_O (min. 98%, VWR BDH Prolabo, Vienna, Austria), CoCl_2_·6H_2_O (min. 98%, VWR BDH Prolabo, Vienna, Austria), in distilled water. Carboxymethyl cellulose was used as surfactant and as precipitant sodium hydroxide was utilized. A stock solution of 1 g Cd (II)/L was prepared from nitrate salt of cadmium [Cd(NO_3_)_2_·4H_2_O]. Distilled water was used to prepare the working solutions by repeatedly diluting the stock solution. Each experiment was run with freshly prepared dilutions. The pH of working solutions was adjusted by adding small volumes of dilute HNO_3_ solution.

### 2.2. Preparation of Magnetic Nanobeads

A one-pot synthesis method is reported in the present study, which can be described as a combination of coprecipitation and crash cooling technology. The process of preparing novel hybrid inorganic nanoparticles CoFe_2_O_4_/CMC (CF@CMC) polymeric framework nanobeads-type adsorbent is summarized in Scheme 1.

The experimental protocol for the synthesis and the details concerning the coprecipitation method have already been reported [57]. Typically, stoichiometric amounts of 42 mL (2.029 g) CoCl_2_·6H_2_O at a concentration of 0.2 M and 42 mL (4.608 g) FeCl_3_·6H_2_O 0.4 M were mixed under vigorous magnetic stirring (800 RPM). The restriction of particle growth was achieved by addition of 84 mL CMC solution (conc. 1%) into the abovementioned mixture.

The mixture was shaken an hour, then an aqueous solution of 3 M NaOH was added dropwise as a precipitant, maintaining the pH value in the range 11–12. After the hydroxides formation, for their conversion into the desired spinel CoFe_2_O_4_ ferrite, a thermal treatment was applied by heating and maintaining the mixture at a constant temperature (80 °C) under a 500 RPM shaking rate for about an hour. The self-assembly process was performed by flash-cooling of the mixture in ice water (from 80 °C to ~0 °C). After the nanobead formation, the samples were magnetic separated and cleaned several times with water followed by drying at 60 °C. The synthetic process described here has major important advantages over previously reported coprecipitation methods since it allows a narrow size distribution, good material yield and fast processing times.

### 2.3. Characterization of Magnetic Nanobeads

The morphology and composition of the samples in the powders state were investigated by using a FIB/FE-SEM CrossBeam 40 NEON EsB (Carl Zeiss, JEOL, Akishima, Tokyo, Japan) equipped with an energy dispersive X-ray spectroscopy (EDS) module. The FE-SEM micrographs were collected at different acceleration voltage and magnifications (1.8 kV, 50 kx; 20 kV, 200 kx; 1.8 kV, 100 kx; and 5 kV, 150 kx). Powder X-ray diffraction analysis (XRD) were performed using an AXS D8-Advance powder X-ray diffractometer (Bruker, Brno, Czech Republic) with CuKa radiation (k = 0.1541 nm). The intensity and voltage of the X-ray source were set at 40 mA and 40 kV, respectively. The samples were scanned in reflection mode in the range 20–80° in 2θ with a step increment of 0.02° per step and a time per step of 0.2 s. Thermogravimetric analysis was performed on a TG/DSC STA 409 PC Luxx Differential Thermal Analyzer instrument (NETZSCH-Gerätebau GmbH–Selb, Germany). The analysis was performed by loading 10 mg of powder samples into an open platinum pan and heated from 30 °C to 800 °C with a rate of 10 °C/min under nitrogen gas flow at around 50 cm^3^/min. The instrument was calibrated for mass loss and temperature using copper sulfate pentahydrate and three-point calibration using lead, aluminium and gold reference standards. The magnetization data were acquired on a model 7410 vibrating sample magnetometer (VSM; Lake Shore Cryotronics, Inc., Westerville, OH, USA).

### 2.4. Adsorption Experiments

All adsorption experiments were conducted at room temperature (23 ± 1 °C), in order to determine optimum parameters by investigating the influence of initial Cd (II) concentration, adsorbent dosage and contact time. For all adsorption experiments reported in this study the pH value was maintained at 5 considering the precipitation issues of Cd(II) ions which tend to precipitate at higher pH values as Cd(OH)^+^ and Cd(OH)_2_ respectively, making adsorption studies impossible. A volume of Cd (II) solution was mixed with intermittent stirring (for a period of 24 h) in a series of centrifuge tubes. The adsorbent was separated from solution by using an external magnet. The supernatant was collected and the determination of Cd (II) was done by spectrophotometric measurements (Perkin Elmer Lambda 35) using xylenol-orange at 576 nm wavelength.

The capacity uptake (q, mg/g) of cadmium ions was determined using the following equation:(1)$$q=\frac{\left(C_{0}-C_{e}\right)V}{m}$$
where C_0_ (mg/L) represent the initial concentration of Cd (II) solution, C_e_ (mg/L) concentration of Cd (II) solution at equilibrium, V (L) volume of solution and m (g) mass of the adsorbent.

#### 2.4.1. Effect of Adsorbent Dosage

The adsorbent dosage was determinate by mixing adsorbent samples ranging between 1.4 and 4.26 g/L at a fixed volume of 10 mL solution/100 mg/L of Cd (II) solution at pH 5.

#### 2.4.2. Effect of Initial Cd (II) Concentration

Isotherms were measured by over the 100–300 mg Cd (II)/L concentration range. For comparison with experimental data two adsorption models were used.

#### 2.4.3. Effect of Contact Time

The optimum time value was determined by conducting adsorption experiments at initial metals ions concentration of 100 mg/L, 1.4 g/L adsorbent dosage at various time intervals between 3 and 120 min.

## 3. Results

### 3.1. Crystal Structure, Morphology and Magnetic Properties of CF@CMC Magnetic Nanobeads

The crystalinity and crystal structure and of CF@CMC magnetic nanobeads were analyzed by powder X-ray diffraction analysis. The XRD pattern (Figure 1) shows that all characteristic diffraction peaks (<220>, <311>, <222>, <400>, <331>, <422>, <333>, <511>, <440>, <531>, <442>, <620>, <533>, <622> and <444> planes), match with the standard CoFe_2_O_4_ JCPDS No.22-1086 thus indicating that the synthesized magnetic nanobeads consist of pure cobalt ferrite phase with no additional observed peaks. In addition, the XRD pattern reveals high background noise and broad diffraction peaks, however, these qualitatively parameters are typical indicative of nanocrystalline samples with very small crystallite size domain (d_XRD_ = 2 nm). Rietveld refinement of CF@CMC nanobeads was employed using the FullProf 2000 program [58,59] by whole profile fitting. The peak shape was modeled by the Thomson Cox Hastings model using a Pseudo-Voigt function [60]. The quantitative parameters were determined by indexing all Braggs reflections in cubic spinel phase with space group Fd-3m and the value of calculated lattice parameter is found to be a = 8.282(3) Å.

The morphology and size of the as-prepared CF@CMC magnetic nanobeads was investigated by SEM analysis. The SEM micrographs of an assembly of CF@CMC magnetic nanobeads are presented in Figure 2.

The SEM micrographs indicate that the as-prepared CF@CMC magnetic nanobeads consist of spherical nanoparticles. The histogram presented as inset in Figure 2 shows that the size varies between 10 nm to 22 nm with a 15 nm mean nanobead diameter.

Thermogravimetric analysis was performed in order to evaluate the stability of the magnetic nanobeads under thermal conditions and to quantify the polymer amounts and is presented in Figure 3.

The TG and DSC curves shows different decomposition steps in the temperature range 25–800 °C. In the temperature range of 25–90 °C the first weight loss assigned to the presence of physically adsorbed water on the CF@CMC nanobeads is observed. Another two thermal events can be observed. The thermal event in the range of temperature between 90–250 °C is accompanied by an exothermic event and a mass loss of approximately 6% correspond to the decomposition of the carboxymethyl cellulose biopolymer. In the last thermal event, which occurs in the 250–800 °C temperature range, a linear mass loss of up to 7.5% can be observed which can be assigned to the removal of extra surface CMC groups. The TGA results indicate that the CMC biopolymer coverage density on the particles is 14% wt. These observations confirmed that the embedded CoFe_2_O_4_ nanoclusters with CMC biopolymer had been formed successfully.

The magnetic properties of CF@CMC magnetic nanobeads are presented in Figure 4.

It is well known that the magnetic properties of CoFe_2_O_4_ are influenced by the composition, crystal structure, crystallite size, cation distributions between octahedral and tetrahedral sites [61,62], surface coatings [63] and synthesis conditions. Cobalt ferrite nanoparticles are ferrimagnetic materials, displaying in some cases a superparamagnetic behavior. For such a ferrite, Chinnasamy et al. suggested a critical single-domain size of CoFe_2_O_4_ of about 40 nm with a coercivity (Hc) of 4.65 kOe [64]. In our case, the bare CoFe_2_O_4_ showed sizes of 2–3 nm and a coercive field of 0.25 kOe, values that are by far more decreased that those generally obtained for this kind of ferrite. Usually, the room temperature coercivity of CoFe_2_O_4_ nanoparticles ranges from 0.5 to 2 kOe [64], but values over 9 kOe have been also obtained [63]. In comparison, bulk cobalt ferrite has a coercivity of about 0.75–0.98 kOe along with 80–90 emu/g specific saturation magnetization (ssM) at room temperature [62,63]. The ssM of bare CoFe_2_O_4_ nanoparticles obtained in this work was 46 emu/g, with a calculated maximum value of 50 emu/g. Therefore, the ssM of the synthesized ferrite is about half that of the bulk counterparts. However, the obtained value of the ssM in this work is high, taking into account that the ssM of the magnetic nanoparticles, including the CoFe_2_O_4_ ones, generally decreases significantly when decreasing the sizes of nanoparticles [65,66].

It can be also noted that the hysteresis loop displayed by the one-pot synthesized cobalt-ferrite nanobeads doesn’t saturate at 20 kOe. The tendency of the ssM to saturate only at extremely high magnetic fields is consistent with other reports on oleic-acid coated CoFe_2_O_4_ and nickel ferrite nanoparticles coated with organic molecules [63,67].

The used polymer hinders sterically the magnetic interaction between nanoparticles inside the nanobeads by fixing them into the matrix. Moreover, the polymer matrix, which contributes with more than 14 percent (*w/w*) to the composition of the nanobeads, probably reduces the surface spin disorders of the nanoparticles, leading to increased ssM. This is contrary to the low ssM obtained for CoFe_2_O_4_ coated by oleic acid [63]. It appears that synthesizing CoFe_2_O_4_ nanoparticles with simultaneously embedding and fixing them into a polymer matrix is much more favorable for an increased saturation magnetization of the final magnetic compound as compared with other approaches that makes use of more fluidic and, therefore, more dynamic organic coatings used for improving different other magnetic properties such as coercively. Finally, despite the high ssM, the relatively low squareness values and coercively of the nanobeads point to a closer superparamagnetic behavior of the as-synthesized magnetic nanobeads.

### 3.2. Influence of Experimental Parameters on Cd (II) Batch Adsorption Experiments

One of the aim of this study was to test the performance of synthesized material for adsorption of Cd (II) ions from aqueous solution. The pH recommended for Cd (II) ions adsorption is 5 because Cd (II) precipitations occurs above pH 6. Therefore, the working pH was set to 5.

#### 3.2.1. Effect of Adsorbent Dosage

The impact of CF@CMC dosage over the adsorption of Cd (II) was studied in a 1.4–4.26 g/L range. The results obtained for the adsorption of Cd (II) by CF@CMC are presented in Figure 5.

The data show that varying the adsorbent dose from 1.4 g/L to 4.26 g/L and keeping constant the other parameters (pH 5, initial concentration of 100 mg/L, temperature 23 ± 1 °C, contact time 24 h) a decrease of Cd (II) capacity uptake from 34.59 mg/g to 19.41 mg/g can be noticed. This variation is due to incomplete usage of CF@CMC adsorption sites and aggregates formation as a response of high quantity of adsorbent. The decrease of Cd (II) adsorption uptake is also mentioned in other studies when the adsorbent dosage is raised, thus 1.4 g/L adsorbent dosage was selected for the rest of the present study.

#### 3.2.2. Effect of Initial Cd (II) Concentration

In order to study the effect of initial concentration, four solutions (100, 150, 200 and 300 mg/L) of Cd (II) were used. The other parameters involved were: pH 5, adsorbent dosage 1.4 g/L, temperature 23 ± 1 °C, contact time 24 h. The effect of the initial Cd (II) concentration of CF@CMC adsorbent is presented in Figure 6.

The maximum capacity uptake of CF@CMC are between 34.59 mg/g and 42.04 mg/g at initial Cd (II) concentrations range of 100–300 mg/L. It can be noted that a maximum of capacity uptake was reached for the 150 mg/L as shown in Figure 6. For higher Cd (II) initial solution concentrations, a plateau is noticed, indicating a constant adsorption capacity as previously reported in literature [68]. One explanation could be that there were no supplementary adsorption sites on the surface of CF@CMC for Cd (II) ions removal.

#### 3.2.3. Effect of Contact Time

It is important to investigate the effect of the contact time required to reach equilibrium. The effect of contact time on the adsorption of Cd (II) at pH 5 with an initial concentration of 100 mg/L, adsorbent dosage 1.4 g/L is shown in Figure 7. The effect of capacity uptake of Cd (II) onto CF@CMC adsorbent was studied at different times ranging from 3–120 min (Figure 7).

As presented in Figure 7, the Cd (II) adsorption on CF@CMC is rapid in the first minutes. The capacity uptake increased from 23.17 to 34.6 mg/g as the contact time increased from 3 to 60 min. The results indicated that after 60 min no further adsorption of Cd (II) was observed. As a consequence, the adsorption equilibrium was set at 60 min from the start of the adsorption process (where the adsorbent was put in contact with the initial solution of Cd (II). It can be emphasized that the fast adsorption presents an advantage for designing water treatment systems for industries.

#### 3.2.4. Adsorption Isotherms

Generally, the adsorption process can be investigated using different isotherm models. However, Langmuir and Freundlich isotherms are the most used models to describe the adsorption processes at equilibrium. Equation (2) describes the linear form of the Langmuir isotherm and Equation (3) describes the linear form of the Freundlich isotherm:(2)$$\frac{C_{e}}{q_{e}}=\frac{1}{K_{L}q_{\max}}+\frac{C_{e}}{q_{\max}}$$
(3)$$\log q_{e}=\left(\frac{1}{n}\right)\log q_{e}+\log K_{F}$$
where: C_e_ (mg/L) is the concentration of metal in solution at equilibrium, q_e_—equilibrium metal adsorption capacity (mg/g); q_max_—maximum adsorption capacity (mg/g); K_L_—Langmuir constant (L/g); K_F_—Freundlich constants (L/g); n—is a constant indicative of adsorption intensity.

The values of the isotherm models are presented in Table 1 along with the determined R^2^ factor. The results indicates that Langmuir model describes better the adsorption process for the CF@CMC-Cd (II) system (Figure 8).

The fundamental characteristic and practicability of Langmuir isotherm regarding a dimensionless constant separation factor or equilibrium parameter, R_L_, is calculated as [69]:(4)$$R_{L}=1/(1+K_{L}C_{0})$$
where: $R_{L}$ is Langmuir constant; C_0_ is the initial concentration of Cd(II).
A value of 0.04 of $R_{L}$ point out that the adsorption is favorable.

#### 3.2.5. Kinetic Studies

In order to study the time related parameters, two models known as pseudo-first-order and pseudo second-order kinetic models were employed [70]. The equation for the pseudo-first-order kinetic model is given by:(5)$$\log\left(q_{e}-q_{t}\right)=\log q_{e}-\frac{\left(k_{1}t\right)}{2.303}$$
where q_t_ is the amount of cadmium adsorbed per unit of adsorbent (mg/g) at time t, k_1_ is the pseudo-first-order rate constant (min^−1^).

The pseudo-second-order model was introduced by Ho and Mckay as described by the following equation:(6)$$\frac{t}{q_{t}}=\frac{1}{k_{2}q_{e}{}^{2}}+\frac{t}{q_{e}}$$
where k_2_ is the pseudo-second-order rate constant (g mg^−1^ min^−1^).

Pseudo-first-order and second order kinetics models plots of Cd (II) adsorption on CF@CMC are showed in Figure 9.

For this particularly case, the pseudo-second order model describes better the adsorptive CF@CMC-Cd (II) system, meaning that the adsorption process is a mono-layered heterogenous one. Kinetic parameters were determined from the slope and intercepts of the linear plots of t/q_t_ against t. The experimental q_e_ value of 35 mg/g is in agreement with the q_e_ value of 35.33 mg/g calculated from the pseudo-second-model and the constant describing the evolution of kinetic system over time (k_2_) has a determined value of 0.0202 g/mg min. With the knowledge that the pseudo-second-order model describes the experimental data can be concluded that the chemical adsorption process is predominant.

The results obtained in this study were compared with some available adsorbents from the literature, Table 2.

## 4. Conclusions

A hybrid inorganic CF@CMC polymeric framework nanobeads-type adsorbent with tailored magnetic properties has been synthesized by a combination of coprecipitation and flash-cooling technology. The prepared material was characterized by means of scanning electron microscopy (SEM), powder X-ray diffraction analysis (XRD), thermogravimetric analysis (TG), and vibrational sample magnetometry (VSM). Precise self-assembly engineering of their shape and composition combined with in depth testing for cadmium removal from wastewater are investigated. The adsorption performance of CF@CMC was evaluated using Cd (II) in an aqueous solution. Initial metal concentration, contact time and adsorbed dosage were the parameters studied for the adsorption experiments performed in this study. The adsorption process is dependent on the adsorbent dosage, initial cadmium concentration, and contact time. The results show that the optimum value of adsorbent dosage is 1.4 g/L. Capacity uptake increases with the increase of initial concentration, reaching the plateau at its maximum of 150 mg/L. In the first minutes the adsorption process is very fast and can be concluded that a maximum of 60 min contact time is highly sufficient to reach the equilibrium state. A Langmuir adsorption isotherm model and pseudo-second-order model describe well the adsorption uptake of the Cd (II) from aqueous solutions by the magnetic nanobeads. The employed models were fitted best with maximum capacity uptake of 44.05 mg/g. The synthesized adsorbent has the benefits of being cheap and easily obtained. Generally, the results show that this type of material can be applied as potential adsorbents for the removal of cadmium ions from aqueous solutions.

## Data Availability

The data presented in this study are available on request from the corresponding author.

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
