# Peer review of "Magnetic Solid-Phase Extraction of Cadmium Ions by Hybrid Self-Assembled Multicore Type Nanobeads"

_polymers, 2021, doi:10.3390/polym13020229_

Round 1

Reviewer 1 Report

In this paper by Buema and colleagues, the authors obtained self-assembled multicore nanobeads and used them for solid-phase extraction of cadmium ions. The results fit the scope of Polymers by providing an interesting solution based on carboxymethyl cellulose. However, at present, the work is not suitable for publication. Considerable corrections should be made to reach the level of scientific rigor expected from a publication in this journal. Please implement the suggestions given below:
1) Please make the description of the novelty factor. It is important to report whether CoFe2O4/CMC materials were produced before, and, if yes, how their properties relate to the findings in this article.
2) Scheme 1 is barely visible. Please increase its size.
3) How was the heating applied? Was it really by a flame underneath as indicated in Scheme 1?
4) Flash cooling procedure is not specified.
5) Characterization parameters such as SEM acceleration voltage or TGA methodology is missing. The list is not exhaustive. I invite the authors to carefully proofread the work and supplement it with a description of relevant parameters.
5) Overall, many experimental details are missing from the manuscript, which makes it irreproducible. Without providing parameters for all the processes, others cannot build on these findings (and verify them), which very much limits the impact of this work.
6) Description of indices in Fig. 1 is barely visible.
7) Professional scale bar markers should be added to Fig. 2 and the redundant information should be removed. The inset is also not legible.
8) The meaning of particles shown in Fig. 3 is unclear.
9) Plots ideally should have the same size and formatting not to confuse the readers.
10) A serious concern is the lack of error bars in Figs. 5-7. Please conduct more experiments, gather enough data, and do the error analysis. At present, there is no evidence to validate if it is justified to interpret these findings.
11) Conclusions section lacks the description of the impact of these results and future outlook.

12) Please consider if citing 7 papers of Buema et al. (first author of this contribution) is absolutely necessary. 

Author Response

Response to comments of Reviewer 1

First of all, we would like to thank you for taking the time to review this article.

On the other hand, we would like to thank you for the comments and suggestions for improving the scientific quality of our manuscript. We have carefully considered the new comments and have revised the manuscript in light of them. The suggested modifications were clearly marked in red in the revised manuscript. Details of our responses to reviewer comments are shown below. We hope you will find these revisions rise to your expectations.

Response to the reviewer's comments point-by-point.

Reviewer #1: In this paper by Buema and colleagues, the authors obtained self-assembled multicore nanobeads and used them for solid-phase extraction of cadmium ions. The results fit the scope of Polymers by providing an interesting solution based on carboxymethyl cellulose. However, at present, the work is not suitable for publication. Considerable corrections should be made to reach the level of scientific rigor expected from a publication in this journal. Please implement the suggestions given below:

1) Please make the description of the novelty factor. It is important to report whether CoFe2O4/CMC materials were produced before, and, if yes, how their properties relate to the findings in this article.

Answer: Thanks for yours suggestions. We added in the introduction part the following paragraf which emphasis the novelty of the present work.

Herein, we have exploited the one-pot experimental method of self-assembled colloidal CoFe2O4 multicore nanobeads using carboxymethyl cellulose (CMC) as polymeric framework. The present synthetic strategy for self-assembling involves (i) the synthesis of CoFe2O4 via coprecipitation solution phase colloidal technique and (ii) the assembly of the magnetic nanobeads by crash-cooling in ice water. This resulted in magnetic nanobeads with a densely packed CoFe2O4 multicore surrounded by CMC polymer shell.

2) Scheme 1 is barely visible. Please increase its size.

Answer: Thank you for your comment. The Scheme 1 was entirely modified as suggested.

3) How was the heating applied? Was it really by a flame underneath as indicated in Scheme 1?

Answer: Thank you for your comments. The thermal treatment was applied using a hot plate and in order to avoid any confusion this now appears much more clearly in the modified Scheme 1.

4) Flash cooling procedure is not specified.

Answer: Thank you for your comment. The mixture of spinel CoFe2O4 ferrite and carboxymethyl cellulose was subjected to a crash-cooling process by placing it in iced water (last stept of syntesis). We observed that the crash-cooling step (from 80°C to ~0°C) leads to heterostructures self-assembled into nonobeads type. A major advantage of the process consists of in its simplicity, reproducibility and high scale of production of the composite material comparing with those reported up to now.

5) Characterization parameters such as SEM acceleration voltage or TGA methodology is missing. The list is not exhaustive. I invite the authors to carefully proofread the work and supplement it with a description of relevant parameters.

Answer: Thanks for yours suggestions. We added in the 2.3 section the following informations.

The morphology and composition of the samples in the powders state were investigated by using a FIB/FE-SEM CrossBeam Carl Zeiss NEON 40 EsB equipped with an energy dispersive X-ray spectroscopy (EDS) module. The FE-SEM micrographs were collected at different acceleration voltage and magnifications (1.8 kV, 50 kx; 20 kV, 200 kx; 1.8 kV, 100 kx; and 5 kV, 150 kx). Powder X-ray diffraction analysis (XRD) were performed using a Brucker AXS D8-Advance powder X-ray diffractometer with CuKa radiation (k = 0.1541 nm). The intensity and voltage of the X-ray source were set at 40mA and 40kV, respectively. The samples were scanned in reflection mode in the range 20-80° in 2θ with a step increment of 0.02° per step and a time per step of 0.2s. Thermogravimetric analysis was performed on Differential Thermal Analyzer TG/DSC NETZSCH STA 409 PC Luxx instrument. The analysis was performed by loading 10 mg of powder samples into an open platinum pan and heated from 30°C to 800°C with a rate of 10°C/min under nitrogen gas flow at around 50cm3/min. The instrument was callibrated for mass loss and temperature using copper sulfate pentahydrate and three-point calibration using lead, aluminium nad gold reference standards.  The magnetization data were acquired on a Lake Shore 7410 vibrating sample magnetometer (VSM).

5) Overall, many experimental details are missing from the manuscript, which makes it irreproducible. Without providing parameters for all the processes, others cannot build on these findings (and verify them), which very much limits the impact of this work.

Answer: Thanks for yours suggestions. We added in the 2.2 section the following informations.

Typically, the stoichiometric amounts of 42 mL (2.029 g) CoCl2·6H2O at a concentration of 0.2 M and 42 mL (4.608 g) FeCl3·6H2O 0.4 M were mixed under vigorous magnetic stirring (800 RPM). The restriction of particle growth was done by addition of 84 mL CMC solution (conc. 1%) into the abovementioned mixture. The mixture was shaken an hour, then aqueous solution of 3 M NaOH as a precipitant agent was added dropwise, maintaining the pH value in the range 11–12. After the hydroxides formation, their conversion into desired spinel CoFe2O4 ferrite, a thermal treatment was applied by heating and maintaining the mixture at constant temperature (80 °C) shaken rate (500 RPM) for about an hour. The self-assembly process was performed by flash-cooling of the mixture in ice water (from 80°C to ~0°C).

6) Description of indices in Fig. 1 is barely visible.

Answer: Thank you for your comment. The Fig 1 was entirely modified as suggested.

7) Professional scale bar markers should be added to Fig. 2 and the redundant information should be removed. The inset is also not legible.

Answer: Thank you for your comment. The Fig 2 was entirely modified as suggested.

8) The meaning of particles shown in Fig. 3 is unclear.

Answer: Thank you for your comment. The main reason behind the presence of particles in the TGA graph is related to the thermal behavior and composition modification of nanobeads upon heating. As stated in the discussions the thermal event in the range of temperature between 90–250 °C correspond to the decomposition of the carboxymethyl cellulose bio-polymer (green shell) depicted in the inset from up right side, and contextually on the last thermal event which occur in the temperature range 250-800° assigned to the removal of extra surficial CMC groups. 

9) Plots ideally should have the same size and formatting not to confuse the readers.

Answer: Thank you for your suggestions. All figures were uniformized.

10) A serious concern is the lack of error bars in Figs. 5-7. Please conduct more experiments, gather enough data, and do the error analysis. At present, there is no evidence to validate if it is justified to interpret these findings.

Answer: Thank you for your suggestions. All figures were now modified and errors bars were added.

11) Conclusions section lacks the description of the impact of these results and future outlook.

Answer: Thank you for your suggestions. The conclusion section was modified as follow:

Hybrid inorganic CF@CMC polymeric framework nanobeads-type adsorbent with tailored magnetic properties is synthesized by a combination of coprecipitation and flash-cooling technology. The prepared material were characterized by means of scanning electron microscopy (SEM), powder X-ray diffraction analysis (XRD), thermogravimetric analysis (TG), and vibrational sample magnetometer (VSM).

Precise self-assembly engineering of their shape and composition combined with deep testing for Cadmium removal from wastewater are investigated. The adsorption performance of CF@CMC was evaluated using Cd (II) in an aqueous solution. Initial metal concentration, contact time and adsorbed dosage were the parameters studied for the adsorption experiments performed in this study. The adsorption process is dependent on the adsorbent dosage, initial cadmium concentration, and contact time.

The results show that the optimum value of adsorbent dosage is 1.4 g/L. Capacity uptake increases with the increase of initial concentration, reaching the plateau at its maximum of 150 mg/L. In the first minutes the adsorption process is very fast and can be concluded that a maximum of 60 minutes contact time is highly sufficient to reach the equilibrium state.

Langmuir adsorption isotherm model and pseudo-second-order model describes well the adsorption uptake of the Cd (II) from aqueous solutions by the magnetic nanobeads. The employed models were fitted best with maximum capacity uptake of 44.05 mg/g.

The adsorbent synthesized have the benefits of being cheap, and easily obtained. Generally, the results show that this type of material can be applied as potential adsorbent for the removal of cadmium ions from aqueous solution.

12) Please consider if citing 7 papers of Buema et al. (first author of this contribution) is absolutely necessary.

Answer: Thank you for your suggestions. The references section was modified as follow:

  • Reference no. 3 was replaced with:

Roman, T.; Asavei, R.L.; Karkalos, N.E.; Roman, C.; Virlan, C.; Cimpoesu, N.; Istrate, B.; Zaharia, M.; Markopoulos, A.P.; Kordatos, K.; et al. Synthesis and adsorption properties of nanocrystalline ferrites for kinetic modeling development. Int. J. Appl. Ceram. Technol. 2019, 16, 693–705.

  • Reference no. 9 was replaced with:

Zhang, W.; An, Y.; Li, S.;  Liu, Z.; Chen, Z.; Ren, Y.; Wang, S.; Zhang, X.; Wang, X. Enhanced heavy metal removal from an aqueous environment using an eco-friendly and sustainable adsorbent. Sci Rep 2020, 10, 16453.

  • Reference no. 10 was replaced with:

Rajczykowski, K.; and Krzysztof Loska, K. Stimulation of Heavy Metal Adsorption Process by Using a Strong Magnetic Field. Water Air Soil Pollut. 2018, 229, 20.

  • Reference no. 69 was replaced with:

Meroufel, B.; Benali, O.; Benyahia, M.; Benmoussa, Y.; Zenasni, M.A. Adsorptive removal of anionic dye from aqueous solutions by Algerian kaolin: Characteristics, isotherm, kinetic and thermodynamic studies. J. Mater. Environ. Sci. 2013, 4, 482-491.

Reviewer 2 Report

Evaluation Report

 This article describes magnetic solid-phase extraction of cadmium ions by hybrid 2 self-assembled multicore type nanobeads. The article is very interesting and well organized with tested results; however, it must be revised in the light of following comments:

  1. Novelty of the work should be emphasized in the introduction part.
  2. The presented results are tests performed on a laboratory scale in very limited conditions of the external environment. In fact the effluents coming out from industries contain several types of ions. So what will be the effect of coexisting ions on the removal of Cd?
  3. pH of solution play a key role in adsorption phenomenon but the authors have only mentioned that removal efficiency is high at pH 5 and has not shown the effect of other pH values. Therefore, in my opinion it is better to show the graph indicating the effect of pH on adsorption.
  4. Experiments describing effect of temperature on the adsorption should be included.
  5. Authors may please insert a Table showing adsorption behavior of Cd towards some other known adsorbents and compare the results of present study with them.

  1. Kinetic study is helpful to calculate the activation energy with Arrhenius equation, so the authors may please calculate the activation energy of adsorption.
  2. Thermodynamic study is very important to determine the spontaneous and non-spontaneous nature of adsorption, but the authors have totally neglected this aspect.
  3. Authors may please remove typo mistakes in the whole manuscript.
  4. What will be the proposed mechanism of adsorption?

Author Response

Response to comments of Reviewer 2

First of all, we would like to thank you for taking the time to review this article.

On the other hand, we would like to thank you for the comments and suggestions for improving the scientific quality of our manuscript. We have carefully considered the new comments and have revised the manuscript in light of them. The suggested modifications were clearly marked in red in the revised manuscript. Details of our responses to reviewer comments are shown below. We hope you will find these revisions rise to your expectations.

Evaluation Report

 This article describes magnetic solid-phase extraction of cadmium ions by hybrid 2 self-assembled multicore type nanobeadsThe article is very interesting and well organized with tested results; however, it must be revised in the light of following comments:

  • Novelty of the work should be emphasized in the introduction part.

Answer: Thanks for your suggestions. We added in the introduction part the following paragraph which emphasis the novelty of the present work.

Herein, we have exploited the one-pot experimental method of self-assembled colloidal CoFe2O4 multicore nanobeads using carboxymethyl cellulose (CMC) as polymeric framework. The present synthetic strategy for self-assembling involves (i) the synthesis of CoFe2O4 via coprecipitation solution phase colloidal technique and (ii) the assembly of the magnetic nanobeads by crash-cooling in ice water. This resulted in magnetic nanobeads with a densely packed CoFe2O4 multicore surrounded by CMC polymer shell.

  • The presented results are tests performed on a laboratory scale in very limited conditions of the external environment. In fact the effluents coming out from industries contain several types of ions. So what will be the effect of coexisting ions on the removal of Cd?

Answer: We totally agree with the Reviewer 2 point of view about the influence of other ions over adsorption capacities. The supplementary researches will be carried in a future study to achieve a more comprehensive evaluation of the adsorbent prepared.

  • pH of solution play a key role in adsorption phenomenon but the authors have only mentioned that removal efficiency is high at pH 5 and has not shown the effect of other pH values. Therefore, in my opinion it is better to show the graph indicating the effect of pH on adsorption.

Answer: In order to avoid precipitation, no higher values were used. The precipitation pHs for Cd2+ occurs at pH around 6.

  • Experiments describing effect of temperature on the adsorption should be included.

Answer: Further work will be focused on the optimization of the process related to thermodynamic study.

  • Authors may please insert a Table showing adsorption behavior of Cd towards some other known adsorbents and compare the results of present study with them.

Answer: The comparison with other materials presented in the specialized literature was added in the manuscript (Table 2)

Table 2. Comparison of maximum Cd (II) capacity uptake (qmax) of different adsorbents reported in literature.

Adsorbent

Capacity uptake, mg/g

References

CF@CMC

44.05

This work

Fe3O4/cyclodextrin polymer nanocomposites (Fe3O4–cyclodextrin)

22.7

[70]

Sulfhydryl functionalized hydrogel with magnetism (Fe3O4–P(Cys/HEA) hydrogel)

19.5

[71]

Amino functionalized magnetic graphenes composite (Fe3O4–GS)

27.8

[72]

Fe3O4 nanoparticles capped with diethyl-4-(4 amino-5-mercapto-4H-1,2,4-triazol3-yl)phenyl phosphonate (DEAMTPP@Fe3O4 MNP)

49.1

[73]

Guanidine-functionalized magnetic Fe3O4 nanoparticles (MNPs-Guanidine)

13.6

[74]

Functionalized magnetic Fe3O4 nanoparticles (Fe3O4@SiO2-NH-py0

45

[75]

Fe3O4/plant polyphenol magnetic material (Fe3O4/PP)

0.951

[76]

  • Kinetic study is helpful to calculate the activation energy with Arrhenius equation, so the authors may please calculate the activation energy of adsorption and Thermodynamic study is very important to determine the spontaneous and non-spontaneous nature of adsorption, but the authors have totally neglected this aspect.

Answer: We thank to Reviewer 2 for this important and useful remark. This research is a preliminary investigation. Further work will be focused on the optimization of the process related to thermodynamic study.

  • Authors may please remove typo mistakes in the whole manuscript.

Answer: Thank you for your observation. We revised the whole manuscript.

  • What will be the proposed mechanism of adsorption?

Answer: The kinetics of cadmium adsorption was described with the Pseudo second order model, which indicated the chemisorption mechanism.

Round 2

Reviewer 1 Report

Thank you. The article was improved substantially. Although the scale bar markers should be enlarged in Fig. 2, it can perhaps be handled at the proof stage. I recommend publication of the article.

Reviewer 2 Report

Please carefully revise the entire text typos and language correction